# A More Rapid Method for Culturing LUHMES-Derived Neurons Provides Greater Cell Numbers and Facilitates Studies of Multiple Viruses

**DOI:** 10.3390/v17071001

**Published:** 2025-07-16

**Authors:** Adam W. Whisnant, Stephanie E. Clark, José Alberto Aguilar-Briseño, Lorellin A. Durnell, Arnhild Grothey, Ann M. Miller, Steven M. Varga, Jeffery L. Meier, Charles Grose, Patrick L. Sinn, Jessica M. Tucker, Caroline C. Friedel, Wendy J. Maury, David H. Price, Lars Dölken

**Affiliations:** 1Institute for Virology and Immunobiology, Julius Maximilian University of Würzburg, Versbacherstr. 7, 97078 Würzburg, Germany; adam.whisnant@uni-wuerzburg.de (A.W.W.); grothey.arnhild@mh-hannover.de (A.G.); 2Institute for Virology, Hannover Medical School, Carl-Neuberg-Str. 1, 30625 Hannover, Germany; 3Department of Biochemistry and Molecular Biology, University of Iowa, 51 Newton Road, 4-403 Bowen Science Building, Iowa City, IA 52242, USA; 4Department of Microbiology and Immunology, University of Iowa, 51 Newton Road, Bowen Science Building, Iowa City, IA 52242, USA; stephanie-e-clark@uiowa.edu (S.E.C.); jose-aguilarbriseno@uiowa.edu (J.A.A.-B.); lorellin_durnell@med.unc.edu (L.A.D.); ann.miller@stjude.org (A.M.M.); patrick-sinn@uiowa.edu (P.L.S.); jmtuckr@uiowa.edu (J.M.T.); wendy-maury@uiowa.edu (W.J.M.); 5Graduate School of Biomedical Sciences, St. Jude Children’s Research Hospital, 262 Danny Thomas Place, Mailstop 1500, Memphis, TN 38105, USA; 6Iowa City Veterans Affairs Health Care System, 601 Highway 6 West, Iowa City, IA 52246-2209, USA; jeffery-meier@uiowa.edu; 7Division of Infectious Diseases, Department of Internal Medicine, University of Iowa Carver College of Medicine, 200 Hawkins Drive, SW54 General Hospital Building, Iowa City, IA 52242, USA; 8Stead Family Department of Pediatrics, University of Iowa, 200 Hawkins Drive, Iowa City, IA 52242, USA; grosec@uiowa.edu; 9Institute for Informatics, Ludwig-Maximilians-Universität München, Oettingenstr. 67, 80538 München, Germany; caroline.friedel@bio.ifi.lmu.de

**Keywords:** LUHMES, neurons, cellular differentiation, herpesvirus, encephalitis, hemorrhagic fever, measles, Zika virus, filovirus

## Abstract

The ability to study mature neuronal cells ex vivo is complicated by their non-dividing nature and difficulty in obtaining large numbers of primary cells from organisms. Thus, numerous transformed progenitor models have been developed that can be routinely cultured, then scaled, and differentiated to mature neurons. In this paper, we present a new method for differentiating one such model, the Lund human mesencephalic (LUHMES) dopaminergic neurons. This method is two days faster than some established protocols, results in nearly five times greater numbers of mature neurons, and involves fewer handling steps that could introduce technical variability. Moreover, it overcomes the problem of cell aggregate formation that commonly impedes high-resolution imaging, cell dissociation, and downstream analysis. While recently established for herpes simplex virus type 1, we demonstrate that LUHMES neurons can facilitate studies of other herpesviruses, as well as RNA viruses associated with childhood encephalitis and hemorrhagic fever. This protocol provides an improvement in the generation of large-scale neuronal cultures, which may be readily applicable to other neuronal 2D cell culture models and provides a system for studying neurotrophic viruses. We named this method the Streamlined Protocol for Enhanced Expansion and Differentiation Yield, or SPEEDY, method.

## 1. Introduction

Neuronal cell culture systems are essential tools for studying neural development, synaptic function, disease mechanisms, and neurotropic viral infections in a controlled environment. These models range from primary neurons, which closely mimic in vivo physiology but pose challenges in scalability and reproducibility, to immortalized neuronal cell lines, which offer convenience but often lack key neuronal characteristics [1]. More recently, conditionally immortalized and stem cell-derived neuronal models, such as induced pluripotent stem cells (iPSCs) and neural progenitor cells, have bridged the gap between physiological relevance and experimental flexibility [2]. Advances in three-dimensional (3D) culture systems, including brain organoids, have further expanded the potential for modeling complex neuronal interactions [3]. The choice of neuronal culture model depends on the specific research question, balancing biological relevance with practicality for mechanistic studies and therapeutic development.

Lund human mesencephalic (LUHMES) cells are a human neuronal precursor line derived from 8-week fetal female mesencephalic tissue and immortalized by a tet-off v-myc oncogene [4]. They are a sub-clone of the MESC2.10 cell line [5]. Upon differentiation, LUHMES cells adopt a mature neuronal phenotype, exhibiting morphological, biochemical, and functional characteristics of dopaminergic neurons. This makes them particularly useful for modeling neurodevelopmental and neurodegenerative processes in vitro. LUHMES maintain a physiologically relevant gene expression profile while offering the scalability and reproducibility required for experimental studies. Their broad applicability spans toxicology [6], cell stress [7], and neurodegenerative disease research [8].

Multiple protocols have been described for differentiating LUHMES cells, with considerable variation between research groups or even within the same group across different papers. In terms of previously published methods, we chose two representative protocols: the one originally employed by Schildknecht et al. [9] and one more recently described by Edwards et al. [10] used for herpes simplex virus research [11]. The Schildknecht protocol involves a pre-differentiation step for two days, followed by detachment of the cells by trypsin and reseeding at a desired density. The Edwards protocol, however, seeds a low density of LUHMES cells, which are allowed to proliferate for 2 to 3 days before differentiation, with no reseeding of the cells. As it had been shown that LUHMES could be directly seeded into the differentiation medium [12], we developed a new method in which high numbers of LUHMES cells are seeded, allowed a recovery time, and differentiated within a few hours after seeding the same day, therefore dubbed the Streamlined Protocol for Enhanced Expansion and Differentiation Yield, or SPEEDY, method.

## 2. Materials and Methods

### 2.1. Cell Culture

Flasks for culturing cells (Corning #430825, Corning, NY, USA) were first treated with poly-L-ornithine (50 µg/mL in water, Sigma #P3655, St. Louis, MO, USA) overnight at room temperature, and then coated with 1 mg/mL fibronectin (Sigma #F2006, New York, NY, USA) in PBS overnight at 37 °C and 5% CO_2_, except for the indicated experiments. Poly-L-lysine (Sigma #P9155, St. Louis, MO, USA) and poly-D-lysine (Sigma #P6407, St. Louis, MO, USA) were used at 100 µg/mL in water, while rat tail collagen I (Corning #354236) was 100 µg/mL in PBS, with the same respective coating times and temperatures as those of ornithine and fibronectin. Flasks were washed three times with water and used immediately or stored at 4 °C up to a week before use. As discussed in the Section 3, drying after coating was unnecessary or even detrimental.

LUHMES cells (ATCC #CRL-2927) were generously provided by David C. Bloom, University of Florida, and maintained at 37 °C and 5% CO_2_. Proliferating LUHMES were maintained in a proliferation medium (DMEM:F12 with L-glutamine, Sigma #D8437) containing 1% N2 supplement (Life Technologies #17502048, Carlsbad, CA, USA), 1× penicillin–streptomycin–glutamine solution (ThermoFisher Sci #10378016, Waltham, MA, USA), and 40 ng/mL recombinant human FGF-basic (Fibroblast Growth Factor, Peprotech #100–18B West Lake Village, CA, USA) and passaged around 75% confluency with 1:4 splits after slight acidity was observed with phenol red. The proliferation medium could be kept as stock without FGF at 4 °C and used within four days after addition. The differentiation medium was defined as the same medium without FGF but with 1 µg/mL tetracycline hydrochloride (Sigma #T7660), 1 mM N6,2′-O-Dibutyryladenosine 3′,5′-cyclic monophosphate sodium (Sigma #D0627), and 2 ng/mL recombinant human Glial cell-derived neurotrophic factor (GDNF, R&D Systems #212-GD-010 Minneapolis, MN, USA). The differentiation medium could be kept several weeks at 4 °C. The culturing materials were purchased from distributors within the USA.

Differentiations are described in detail in the Section and corresponding figures. For the Schildknecht method, an equivalent of 8 × 10^6^ cells in a T150 were seeded in the proliferation medium. The differentiation medium was added the next day, for 2 days total before enzymatic detachment. The pre-differentiated cells were then re-seeded in the differentiation medium at an equivalent of 2.4 × 10^7^ cells per T150 for an additional 3 days. For the Edwards method, multiple reported cell numbers and times were tested. An equivalent of 2 or 3.1 million cells per T150 were seeded for two or three days in the proliferation medium. This medium was then exchanged for the differentiation medium, and again for a fresh differentiation medium two days later. Aside from the different cell numbers tested in Figure A1C, the SPEEDY method involved seeding 2.5 × 10^7^ cells per T150 in the proliferation medium, followed by an exchange to the differentiation medium after four hours’ recovery. This medium was replaced with a fresh differentiation medium again after two days, as some slight acidity with phenol red was observed. After the exchange on day two of differentiation, neurons were maintained for all protocols with a fresh differentiation medium every three days without noticeable changes in pH.

### 2.2. Viruses

Recombinant VSV viruses pseudotyped with hemorrhagic fever virus glycoproteins have been previously described for Ebola [13], Lassa [14], and Marburg [15]. GFP-expressing measles [16], MHV68 [17], and HSV-1 [18] were grown as previously described. Zika virus (clinical isolate FSS13025) expressing a Renilla luciferase reporter has been previously described [19].

### 2.3. Immunofluorescence

Cells were plated and differentiated on coated German glass coverslips (Electron Microscopy Sciences #72290-04, Hatfield, PA, USA), with 5 × 10^5^ cells per well of a 12-well dish. Infections were carried out as described in the relevant figures before fixation with 4% formaldehyde in 1x PBS for 20 min at room temperature and washed once with PBS. Cells were then stained using a solution of 10% FBS, 1x PBS, 0.1% saponin, and β3 Tubulin antibody 2G10 (Santa Cruz Biotechnology #sc-80005, 2 µg/mL, Dallas, TX, USA) overnight at 4 °C. Cells were washed once more with PBS and stained in the same buffer but using Goat anti-Mouse IgG (H+L) Cross-Adsorbed Secondary Antibody Alexa Fluor™ 647 (ThermoFisher #A-2123) at a 1:1000 dilution for one hour at room temperature. DAPI staining was conducted at 0.5 µg/mL for 5 min in water at room temperature before mounting overnight at 4 °C with ProLong™ Diamond Antifade Mountant (ThermoFisher #P36961 Waltham, MA, USA). Imaging was performed on a Leica DMR epifluorescent scope fitted with a SPOT RT sCMOS camera. Images were acquired using the SPOT software and processed in Fiji. Live cell imaging was performed on a Leica DMi8. The staining materials were purchased from distributors within the USA.

### 2.4. qRT-PCR

RNA was extracted using TRIzol and 1 μg converted to cDNA in 20 µL using random hexamers using the High-Capacity cDNA Reverse Transcription Kit (Applied Biosystems #4368814, Waltham, MA, USA). qPCR was performed using Power SYBR Green Master Mix (Applied Biosystems #4368708, USA) and the following primers listed 5′-3′: CAAGGCCTTTAGTCCCACAATTC and GATGACCCCAATGTCTGAACGATC for VSV-L, GTCGGAGGTAACGGACGAATG and GAATGGATTGGGATAACACTTAGATTG for VSV-P, and GGTGTCAACCATGAGAAGTATGA and GAGTCCTTCCACGATACCAAA for GAPDH.

### 2.5. Luciferase Assay

LUHMES were seeded 5 × 10^4^ cells per well in coated 96-well plates. Cells were infected with Zika at the MOIs described. The experiments were performed in triplicate with 3-fold MOI dilutions in the range of 0.001–10 starting from a stock of 3.16 × 10^9^ IU/mL. For inhibitor studies, the infected cells were incubated with Bemcentinib (BGB324, R428) in 2-fold dilutions in the range of 0.5–1.5 µM. After 48 h, the medium was removed, and cells were gently washed with PBS. Luciferase expression was measured by a Promega Renilla Luciferase Assay System Kit (#E2810), according to manufacturer’s instructions. Relative light units (RLUs) were measured by a BioTek Synergy H1 plate reader. Triplicates of infected wells were averaged and subtracted from mock uninfected wells.

### 2.6. RNA Sequencing and Data Analysis

In this step, 5 × 10^5^ cells were seeded in 6-well plates and differentiated according to the SPEEDY protocol or collected as undifferentiated cells. Cells were lysed in TRIzol and RNA collected with the Direct-zol RNA Miniprep kit (Zymo Research #R2050, Irvine, CA, USA) and submitted to BGI (Hong Kong) for processing. Briefly, total RNA was treated with DNase I and then Ribo-Zero before fragmentation to 130–160 nt. cDNA synthesis was followed by sequential processing with End Repair Mix and A-Tailing Mix. The adenylated 3′-ends DNA were ligated to the RNA Index Adapter using Ligation Mix before Ampure XP Beads purification. The double-stranded PCR products were heat-denatured and circularized by the splint oligo sequence. The single-stranded circular DNA (ssCir DNA) was formatted as the final library. The library was amplified with phi29 to make DNA nanoballs (DNBs), which have more than 300 copies of one molecular template. The DNBs were loaded into the patterned nanoarray and 100 base paired-end reads were generated in the way of combinatorial Probe-Anchor Synthesis (cPAS) and sequenced on a DNBSeq-G400.

The quality of the sequencing data was checked using FastQC (version 0.11.9) [20]. RNA-seq reads were mapped against the (i) the human genome (GRCh38/hg38), (ii) human rRNA sequences, and (iii) the HSV-1 genome (HSV-1 strain 17, GenBank accession code: JN555585), using ContextMap 2 (version 2.7.9) [21] and using BWA as short read aligner and with default parameters [22]. SAM output files of ContextMap2 were converted to BAM files using samtools [23]. The number of read fragments per gene was determined from BAM files in a strand-specific manner using featureCounts [24] and gene annotations from Ensembl (version 100). All these steps were implemented and run using the Watchdog workflow management system [25]. Log2 fold-changes between differentiated and undifferentiated cells were determined with DESeq2 [26]. Statistical analysis was calculated in GraphPad Prism 8.

## 3. Results

### 3.1. Evaluation of Three LUHMES Differentiation Methods

Variability in the number of viable cells sometimes occurred after trypsinization when using the Schildknecht method, while the Edwards protocol resulted in cellular aggregates and empty spaces within the culture. We thus sought to evaluate what methods reliably produced the greatest number of evenly dispersed, mature neurons. In general, initiating neuronal differentiation immediately after splitting resulted in fewer cell numbers through a combination of detachment and cell death during the 5 days of differentiation and reduced cell division during the first two days after the removal of the proliferation medium. On the contrary, allowing the cells to recover a day or more in the proliferation medium allowed the undifferentiated cells to migrate towards each other and generate cell aggregates. We therefore aimed to resolve both issues by seeding a large number of cells and initiating differentiation after a brief recovery period, before the cells began to aggregate, and only exposing them to one round of trypsinization. As we desired to optimize conditions for large-scale neuronal culturing, the following cell numbers are for T150 flask equivalents. However, these methods are scalable down to 96-well plates using the equivalent cell number to surface area ratio.

To evaluate the Schildknecht method of pre-differentiation and reseeding, 5 × 10^5^ cells were seeded overnight per well of a 6-well dish (roughly equivalent to 8 × 10^6^ cells in a T150) in the proliferation medium, followed by differentiation for 2 days before enzymatic detachment. The pre-differentiated cells were then re-seeded at an equivalent of 2.4 × 10^7^ cells per T150 for an additional 3 days. This protocol, therefore, takes 6 days in total. As shown in Figure 1A, this protocol results in a relatively high and even distribution throughout the culture. This protocol was compatible with multiple detachment reagents, including 0.25% trypsin-EDTA, TrypLE, and Accutase (Appendix A Figure A1A). Each of these solutions generally took the same amount of time for cellular dissociation. When using this protocol, higher numbers of viable cells were obtained when letting the cells detach in the solution for longer periods of time than attempting to accelerate the process by forceful pipetting. The main disadvantage of this protocol is variability in the overall number of viable cells available for reseeding after detachment and resuspension.

Another method of culturing LUHMES, recently used by Edwards and colleagues, involves the seeding of cells and proliferation for 2 or 3 days followed by 5 days of differentiation, therefore taking 7 to 8 days in total. Numbers of cells matching publications using this method were allowed to proliferate before differentiating. As LUHMES tend to aggregate [27], even after attachment as single cells, this protocol results in areas where cell bodies are concentrated in centers with axonal outgrowths then spreading across as shown in Figure 1B, Figure A1B and Figure A2. In addition to being the longest of the published protocols, this method results in a variable density of the neurons within the culture.

To achieve a high number of mature neurons with a more even distribution across the culture, we tested if they could be seeded directly and if differentiation induced on the same day minimized grouping resulting from proliferation and migration. In general, cells stably reattached to the cell culture dish within 3–4 h after splitting. We thus tested whether the two respective problems of cell loss from differentiating too soon after splitting vs. cellular aggregation could both be solved by briefly allowing the undifferentiated cells to reattach for 4 h in the proliferation medium before changing to the differentiation medium. As shown in Figure 1C, this method resulted in a relatively high density of neurons with evenly distributed cell bodies, with a similar distance between nuclei as seeding pre-differentiated neurons via the Schildknecht method (Figure A2). We tested our approach for a range of cell numbers spanning 5–30 million per T150 flask, all of which reliably resulted in mature neuronal morphology (Figure A1C). The four-hour recovery was found to be beneficial to the cells compared to seeding directly in the differentiation medium (Figure A3).

The maximal density was achieved with seeding 2.5 × 10^7^ cells per T150 flask. As in the range of 5–5.5 × 10^7^ the cells could be gathered from the proliferating T150 cultures before seeding, this number was also convenient as it routinely allows two flasks for differentiation from each flask of ~80% confluent LUHMES. Overall, this protocol takes 5 days and results in the greatest achievable density and numbers of mature neurons without the problem of cell aggregate formation.

### 3.2. Transcriptome Analysis of LUHMES Differentiation

While our new method resulted in mature neuronal morphology, we sought to verify that the gene expression profiles matched those expected from previous studies, particularly regarding changes in the transcriptome when changing from proliferating to differentiated states. We thus performed RNA-Seq on neurons generated via the SPEEDY method and compared these to previously published data from Pierce et al., which utilized the original Schildknecht method [28]. Integrating the two datasets, there were 16,167 genes with mapped reads in both studies.

The comparison of the transcriptomes of undifferentiated and differentiated LUHMES revealed very similar changes for each gene during differentiation with the SPEEDY method to those previously reported (Pearson r = 0.9095, *p* < 0.0001, Figure 2A). We then performed a principal component analysis on the overall gene expression profiles of undifferentiated and differentiated LUHMES. Interestingly, the first principal component in the principal component analysis (PCA, Figure 2B) separated undifferentiated from differentiated cells from both studies, while only the second component separated the two studies. Thus, most of the variance is explained by the difference between undifferentiated and differentiated cells. Overall, we conclude that the SPEEDY method results in mature neurons matching the expected gene expression patterns.

### 3.3. Evaluation of Coating Conditions for the SPEEDY Method

Multiple flask coating conditions have been reported in the literature for both LUHMES and their parental cell line MESC2. These include differences in both the cationic polymers for the first coating (commonly poly-L-ornithine but also poly-L-lysine [29] or poly-D-lysine [12]) and the subsequent protein coating with fibronectin, collagen, or laminin [5]. Matrigel has also been used for 3D cultures of LUHMES neurons [30,31], while PEGylated polymers facilitate microfluidic culturing [32]. We thus decided to test combinations of the more common coating reagents for their compatibility with our new method. As shown in Figure 3, all combinations of poly-L-ornithine and -L- or -D-lysine were compatible with fibronectin or collagen coating to generate mature neurons via the SPEEDY method. Furthermore, distances between nuclei were generally comparable between coating methods (Figure A4). As the combination of ornithine and fibronectin has been more recently favored in the literature, we thus continued this as the default coating condition for the following studies.

Different groups have commented that LUHMES detach after 9–12 days of differentiation [27,31], while others have published experiments of 20 days for studies involving herpesviral latency [10]. This suggests that variations in protocols between labs can change the stability of cultures by weeks. During the course of this work, we found two handling steps that greatly impacted the longevity of LUHMES cultures. The primary recommendation for preventing detachment is not drying the flasks after coating, as over-drying caused a more rapid loss of cultures. After washing with water to remove unbound fibronectin or collagen, simply refrigerating the plastic or glass in a sealed chamber to keep residual moisture allowed pre-coating for at least a week in advance. The second recommendation is, after one medium exchange at 2 days of differentiation, to change medium every 3 days. This has been reported previously and decreases the likelihood of monolayer detachment due to handling, especially over long-term cultures.

### 3.4. High-Density LUHMES Cultures as a Model for Multiple Viruses

#### 3.4.1. Entry via Glycoproteins of Hemorrhagic Fever Viruses

As our new method provides substantially more neurons per culture than previous protocols, we sought to test whether high-density differentiated LUHMES cells could be used to study rare events in neuronal infection biology. Hemorrhagic fever (HF) associated with viral infection is often associated with a range of neurological disorders, including delirium, seizures, and coma. However, it is difficult to assess neuropathic contributions from the direct infection of brain cells versus systemic issues, such as inflammation and blood–brain barrier breakdown. To determine if LUHMES neurons could indeed be infected by HF viruses, we employed a reporter system based upon vesicular stomatitis virus (VSV) in which the native VSV-G glycoprotein is replaced with glycoproteins of viruses of interest. As VSV contains a single negative-sense RNA genome, the detection of viral mRNA by either a reporter gene expression or RT-PCR can be used as a readout of infection.

As shown in Figure 4, a recombinant VSV (rVSV) containing the Ebola virus glycoprotein (EBOV GP) could enter differentiated LUHMES as determined by GFP expression. Entry was roughly 100-fold less efficient than the same rVSV containing the VSV-G glycoprotein (Figure 4A vs. Figure 4B). Non-fluorescent rVSV with the glycoproteins of either the Lassa or Marburg virus were also tested for entry, and viral gene expression was determined by qRT-PCR for the P and L mRNAs. As shown in Figure 4C,D, both recombinant viruses could enter LUHMES and begin transcription in dose-dependent manners. However, the Lassa virus GP appeared to facilitate entry far more efficiently as amplification values were detectable at 10,000-fold lower concentrations of virus. Our data demonstrate that, although with differing efficiency, LUHMES neurons can facilitate viral entry via the receptor glycoproteins of different hemorrhagic fever viruses, including filoviruses (Ebola and Marburg) and arenaviruses (Lassa). The higher efficiency of Lassa entry is likely due to the higher expression of α-dystroglycan (DAG1) than of Tyro3 protein kinase DTK (Table A1). DAG1 was previously found in mature LUHMES by mass spectrometry [12], while Tyro3 was the only receptor found in our sequencing to support entry with Marburg and Ebola receptors [33], but can also support Lassa [14].

#### 3.4.2. Childhood Encephalitis Viruses

There are relatively few well-established models for studying measles virus (MeV)-induced encephalitis, and each comes with limitations. Primary human neurons and transgenic animal models provide the most physiologically relevant systems, but are difficult to culture and maintain. LUHMES cells and other neural progenitor-derived models offer an alternative with greater accessibility and consistency, though they may not fully recapitulate complex brain environments. The ability of LUHMES-derived neurons to be cultured on the scale of months also allows for longer-term studies of viral reactivation, which can occur years after primary infection in vivo.

To evaluate the ability of high-density LUHMES cultures as models for the neuronal infection of encephalitis viruses, we infected cells with a GFP-expressing measles virus at a range of different MOIs. As shown in Figure 5A, GFP-expressing cells demonstrated successful virus entry and gene expression. Interestingly, measles appeared to be incredibly cytopathic at higher MOIs as shown by the large areas of cellular detachment. These data demonstrate the ability of LUHMES cells to study neuronal infection and inflammation caused by the measles virus. We could not detect the expression of known measles virus receptors CD150/SLAMF1 or Nectin 4/PVRL4 by RNA-Seq, but potential entry mechanisms are elaborated upon in the Section 4.

Undifferentiated LUHMES cells have previously been shown to be infected by the Zika virus [34], which caused widespread encephalopathy in the 2015–16 outbreak. We tested whether the new differentiation protocol was also suitable for Zika infection using a luciferase-expressing virus. As shown in Figure 5B, we could detect viral gene expression in an MOI-dependent manner, validating the system as an infection model in both differentiated and progenitor states. Entry is likely through the Tyro3 receptor (Table A1), as we were able to map very few RNA-Seq reads to the AXL tyrosine kinase. However, an AXL inhibitor Bemcentinib (BGB324, R428) that was previously shown to inhibit Zika [35] also lowered the infection of LUHMES at 1 μM (Figure 5C). While Bemcentinib has greater specificity to AXL, it was shown to have some inhibitory activity against Tyro3 [36]. No evidence of infection was observed even up to MOI 30 for a fluorescent respiratory syncytial virus (RSV).

#### 3.4.3. Herpesviruses

LUHMES neurons have recently been utilized by multiple groups to investigate both lytic and latent infection by herpes simplex virus type 1 (HSV-1) [10,11]. In regard to latency, they provide a major advantage to other models in the ability to maintain latent viral genomes in the absence of acyclovir [37]. Neurons differentiated by both the Edwards and SPEEDY methods showed comparable infection rates using a GFP-expressing HSV-1 at 8 h post-infection with an MOI of 10 HFF units (Figure 6A), indicating that HSV entry and kinetics are comparable across protocols.

We also sought to investigate other herpesviruses, including the betaherpesvirus human cytomegalovirus (HCMV) and gamma herpesvirus murine herpesvirus 68 (MHV68). As with RSV, we saw no evidence of viral gene expression with HCMV. In contrast, a GFP-expressing MHV68 was able to enter LUHMES and express viral genes (Figure 6B). Though a murine virus, MHV68 is genetically and biologically similar to human gammaherpesviruses such as KSHV and EBV, sharing conserved genes and replication strategies, and is a tractable in vivo model to study gammaherpesvirus pathogenesis [38]. LUHMES cells may provide a more facile model to study the fatal central nervous system infections of MHV68 described in mice [39,40].

## 4. Discussion

In this paper, we describe a new method of differentiating LUHMES neurons that provides a greater number of cells at an even distribution without cell aggregate formation in a shorter time frame. The SPEEDY method takes 5 days and reliably generates predictable cell cultures from the proliferating progenitor cells with fewer handling steps that can introduce variability between experiments or the loss of cells when planning large-scale experiments. This method provides advantages in the loads of neuronal cells for high-throughput studies and facilitates the capture of rare events that would require greater numbers of individual cultures. This method was also robust with multiple coating conditions that have been reported in the literature for other differentiation protocols. The reduced clumping of cells allows for a more homogenous exposure to viruses during infection and facilitates resuspension for single-cell assays, such as flow cytometry and RNA-Seq.

SPEEDY LUHMES cultures provide a suitable model to investigate the neuropathy of multiple viral families. In this study, we show that the glycoproteins of filoviruses Ebola and Marburg, as well as the Lassa arenavirus, can enter neuronal cells. While we relied on VSV reporter viruses due to biosafety considerations, we provide a basis for future studies that can employ native viruses to study downstream events. LUHMES neurons were also infectable by the measles virus, which caused a significant cytopathic effect, indicating that direct neuronal infection may contribute significantly to brain inflammation in vivo. Despite the absence of SLAMF1 and NECTIN4 receptors, LUHMES being permissive to wild-type measles virus suggests the engagement of noncanonical entry or spread pathways. A recent work by Yousaf et al. demonstrated that a single matrix protein mutation, M-F50S, confers receptor-independent spread in human iPSC-derived neurons, bypassing the need for SLAM or NECTIN4 [41]. Importantly, this receptor-independent fusion relies on interactions with the “cis-acting receptors” CADM1 and CADM2. Because LUHMES express CADM1/2 and NECTIN1/PVRL1 (Table A1), it is plausible that the measles virus similarly exploits these cell-surface proteins to achieve infection in LUHMES.

Multiple groups have employed LUHMES to study herpesviruses, particularly alphaherpesvirus HSV-1, and we demonstrate that they can also be infected by gammaherpesvirus MHV68. Our new protocol is also able to support the infection of HSV-1 and the Zika virus, as has been described with other methods in LUHMES. While we saw no direct viral gene expression with RSV or HCMV, we did not evaluate if the viruses are incapable of entering cells or have other blocks post-entry.

A major limitation of this study is that we focused primarily on cellular morphology, RNA profiles, and viral infection as biological readouts, rather than evaluating neuronal phenotypes, such as dopamine production or synaptic signaling. Such measurements are limited in the literature, or even demonstrate conflicting results depending on the source of the cells [42]. While these readouts are certainly important for understanding neuronal biology and can hopefully be addressed in future studies, we focus on the ability of our new method to generate large numbers of neurons for infection biology.

## Figures and Tables

**Figure 1 viruses-17-01001-f001:**
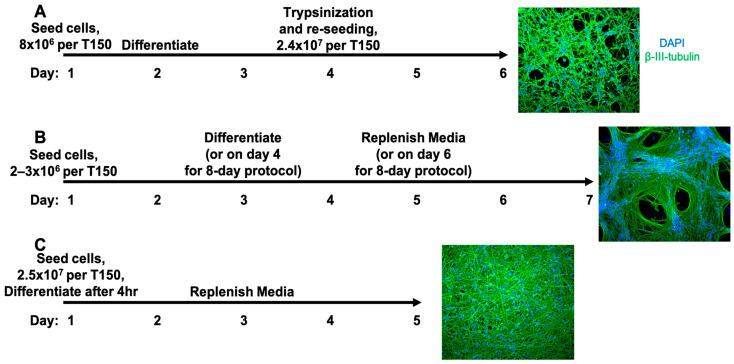
Immunofluorescence visualization of culturing methods for LUHMES-derived neurons after 5 days of differentiation. (**A**) The method described by Schildknecht et al. involving pre-differentiation for two days, followed by trypsinization and reseeding at a high density. (**B**) The method published by Edwards et al., allowing small numbers of cells to proliferate before differentiation. (**C**) The SPEEDY method of LUHMES cells where high numbers of cells are allowed to attach and recover from trypsinization in the proliferation medium for four hours, before inducing differentiation.

**Figure 2 viruses-17-01001-f002:**
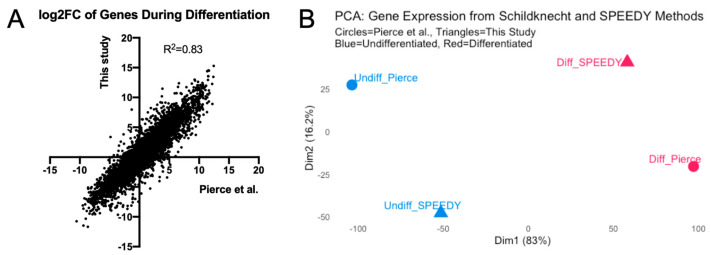
Transcriptional similarity of LUHMES differentiated by the SPEEDY and Schildknecht methods. (**A**) The log2 fold-changes (FCs) of genes during differentiation were determined by RNA-Seq. The values from our study with the SPEEDY differentiation protocol were plotted on the y-axis, while previously published values using the Schildknecht method [28] were plotted on the x-axis. (**B**) Principal component analysis of the overall gene expression of undifferentiated (blue) and differentiated (red) LUHMES generated by the SPEEDY method (triangles) and Schildknecht method (circles).

**Figure 3 viruses-17-01001-f003:**
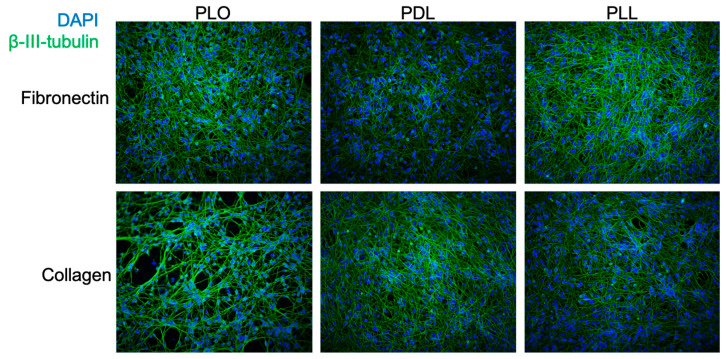
SPEEDY LUHMES culturing is compatible with multiple coating strategies. Coverslips were coated with poly-L-ornithine (PLO), poly-L-lysine (PLL), and poly-D-lysine (PDL), followed by coating with fibronectin or collagen before cell seeding and differentiation.

**Figure 4 viruses-17-01001-f004:**
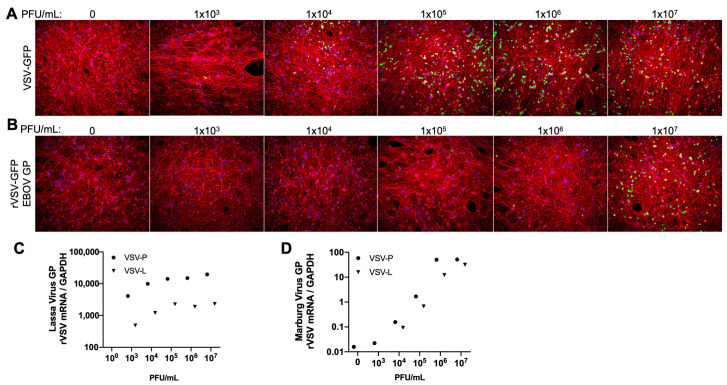
High-density LUHMES cultures can be infected using envelope proteins from viruses causing hemorrhagic fever. Infection rates of a recombinant vesicular stomatitis virus (rVSV) expressing GFP, which was used to visualize viral entry and gene expression with virions containing the (**A**) VSV-G envelope protein and (**B**) Ebola virus glycoprotein (EBOV GP). DAPI signal is in the blue channel, while red is β-III tubulin staining. A non-fluorescent rVSV virus expresses the P and L mRNAs as measured by qRT-PCR with GP envelopes from the (**C**) Lassa virus and (**D**) Marburg virus. Samples were harvested after 24 h of infection.

**Figure 5 viruses-17-01001-f005:**
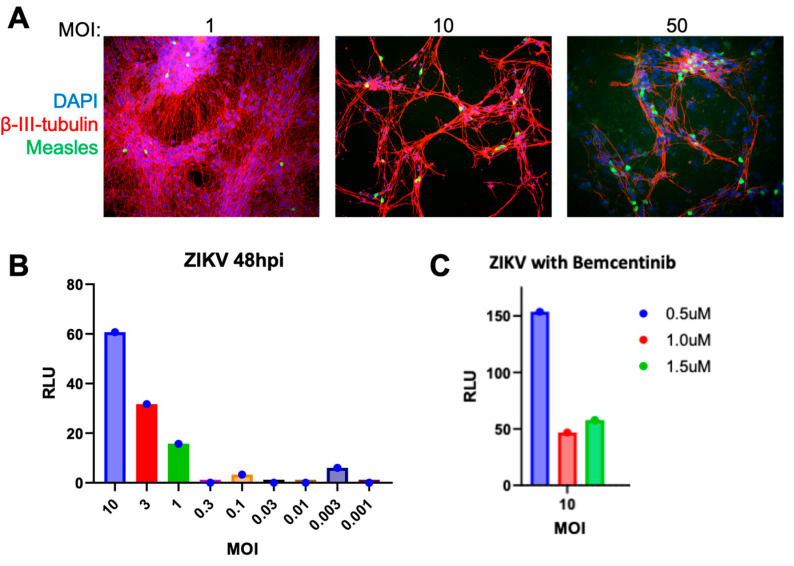
LUHMES neurons can be infected with viruses associated with childhood encephalitis. High-density LUHMES cultures were infected with a range of MOIs for (**A**) GFP-expressing measles virus at 24 h and (**B**) a luciferase-expressing Zika virus at 48 h post-infection (hpi). (**C**) Zika virus (ZIKV) was infected at MOI 10 in the presence of increasing amounts of Bemcentinib and relative viral gene expression measured assayed by luciferase activity.

**Figure 6 viruses-17-01001-f006:**
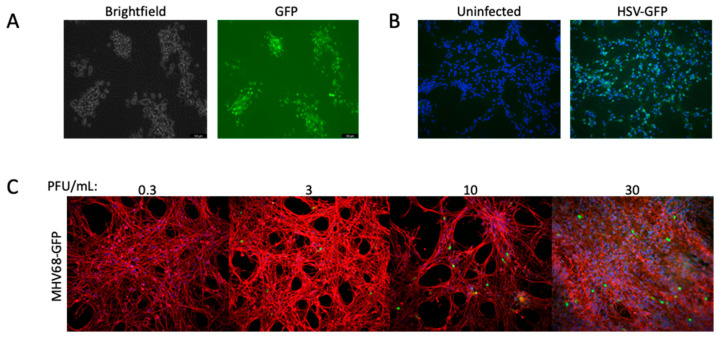
LUHMES neurons can support studies of multiple herpesviruses. Differentiated neurons were infected with: (**A**) a GFP-expressing HSV-1 at MOI 10 for 8 h with the Edwards (live-cell imaging) and (**B**) SPEEDY (immunofluorescence) methods, or (**C**) a GFP-expressing murine herpesvirus 68 at varying MOIs for 24 h.

## Data Availability

The RNA-Seq data are available on the Gene Expression Omnibus (GEO) under accession record GSE297466.

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
