# Peer review of "A More Rapid Method for Culturing LUHMES-Derived Neurons Provides Greater Cell Numbers and Facilitates Studies of Multiple Viruses"

_viruses, 2025, doi:10.3390/v17071001_

Round 1

Reviewer 1 Report

Comments and Suggestions for Authors

The article is interesting, innovative, and well-written. However, I have a number of concerns that prevent me from recommending the article for publication.
1) The study focuses on morphology and viral infection but does not include functional validation (e.g. electrophysiology, dopamine secretion, or synaptic activity). This calls into question whether SPEEDY neurons truly recapitulate mature phenotypes. The authors should consider using an additional method
2) There are no data to confirm whether HCMV/RSV entry is blocked or whether there is a post-entry failure.
3) The statement about the absence of aggregates in SPEEDY cultures contradicts Fig. 1C, where cellular clumps are visible. There is no quantification of aggregation frequency.
4) Fig. 3 does not include statistical analysis of neuronal yield or survival under different conditions. Data on aggregate size/frequency and statistical comparison between protocols should be provided.

5) Long term stability of cultures (>12 days) is mentioned but not tested.
6) Table A1: Add statistical significance of differences in receptor expression between studies.

Reviewer 2 Report

Comments and Suggestions for Authors

In the manuscript submitted to me for review entitled "A More Rapid Method for Culturing LUHMES-Derived Neurons Provides Greater Cell Numbers and Facilitates Studies of Multiple Viruses“ the authors Adam W. Whisnant, Stephanie E. Clark, José Alberto Aguilar-Briseño, Lorellin A. Durnell, Arnhild Grothey, Ann M. Miller, Steven M. Varga, Jeffery L. Meier, Charles Grose, Patrick L. Sinn, Jessica M. Tucker, Caroline C. Friedel, Wendy J. Maury, David H. Price and Lars Dölken present the development of a new method for the differentiation of a model of Lund human mesencephalic (LUHMES) dopaminergic neurons, which is faster than other protocols and leads to the production of a larger number of mature neurons. The authors also show that the resulting neuronal culture may provide a system for studying neurotrophic viruses.

My remarks and recommendations to the authors are:

  1. On line 76 it is stated:

"in PBS overnight at 37°C"

Was it incubated in the presence of CO2 or not? If "YES", it is good to state this in the text. It is also not stated in the other incubations. If it was incubated in the presence of CO2 somewhere, it is also good to state this.

  1. In the Materials and Methods section, the city and country of the manufacturing companies are not stated. It is good to supplement the missing information.
  2. Have the specified viruses been replicated previously in another type of cell culture?
  3. On lines 142-143 it is stated:

"Experiments were performed in triplicate with 3-fold MOI dilutions ranging from 0.001-10".

In order to determine the MOI of the viral infection, this means that the initial titer of the virus must be known. However, the titer of the virus is not stated anywhere in the text.

  1. In the References section, many of the references list the first author and indicate "et al.,". This is allowed when there are more than 10 authors. Please add the missing authors. See instructions for authors.

For documents co-authored by a large number of persons (more than 10 authors), you can cite the first ten authors, then add a semicolon and add ‘et al.’ at the end:

Author 1; Author 2; Author 3; Author 4; Author 5; Author 6; Author 7; Author 8; Author 9; Author 10; et al.

Round 2

Reviewer 1 Report

Comments and Suggestions for Authors

The file with the answers to my comments is damaged and I cannot study the authors' answers. Please upload the file with the answers.

Author Response

Please see the attachment for data figures, but here is the text:

We thank the reviewers for their thoughtful and constructive feedback, which has helped us improve the clarity and rigor of our manuscript. We have addressed each comment carefully and revised the text, figures, and appendix materials accordingly. Major changes include the addition of new data on cellular aggregation, clarification of experimental conditions and viral titers, expanded text discussion, and corrections to reference formatting. We believe these revisions have strengthened the manuscript and clarified the scope and interpretation of our findings. A point-by-point response is below, with our comments highlighted. 

Reviewer 1: 

Comments and Suggestions for Authors 

The article is interesting, innovative, and well-written. However, I have a number of concerns that prevent me from recommending the article for publication. 

  1. The study focuses on morphology and viral infection but does not include functional validation (e.g. electrophysiology, dopamine secretion, or synaptic activity). This calls into question whether SPEEDY neurons truly recapitulate mature phenotypes. The authors should consider using an additional method 

We thank the reviewer for this important point and fully agree that functional validation is critical for confirming neuronal maturity. While we have initiated project discussions with neuroanatomy researchers at our institute, these assays require substantial setup time and resources. In the meantime, we hope the combination of morphological and transcriptomic analyses, each consistent with prior LUHMES studies, provides sufficient evidence for the neuronal identity of our cultures until we are able to establish the above assays, particularly regarding conflicting results in the literature that are discussed in the text. 

  1. There are no data to confirm whether HCMV/RSV entry is blocked or whether there is a post-entry failure. 

We agree this is an important question. For HCMV, our qPCR data suggest that viral DNA may enter the cell and initiate limited gene expression, although we observe only sporadic expression of fluorescent proteins in LUHMES neurons. These findings suggest a post-entry block, but definitive conclusions require further experiments. We plan to pursue follow-up studies using methods such as super-resolution microscopy, RNA-Seq, and ChIP to delineate the stage at which these viruses are restricted. 

  1. The statement about the absence of aggregates in SPEEDY cultures contradicts Fig. 1C, where cellular clumps are visible. There is no quantification of aggregation frequency. 

Thank you for this observation. We have now included a quantification of clumping as Appendix Figure A3 and have expanded the relevant discussion in the main text to clarify our terminology and address the presence of cell clusters. 

  1. Fig. 3 does not include statistical analysis of neuronal yield or survival under different conditions. Data on aggregate size/frequency and statistical comparison between protocols should be provided. 

We agree that quantitative assessment of viability would strengthen the figure. While each coating condition is supported by prior studies and yields cultures with comparable morphology and density, we now include measurements of aggregate size in Appendix Figure A4 and discuss this in the main text. These data provide further insight into the relative efficiency of the protocols. 

  1. Long term stability of cultures (>12 days) is mentioned but not tested. 

We appreciate this point. Although we have cultured LUHMES-derived neurons for more than three weeks in the context of HSV latency pilot studies, we did not acquire publication-quality images of those cultures. We now reference relevant studies from the Bloom lab demonstrating LUHMES culture for at least 20 days, and we have included this citation and clarification in the text. 

  1. Table A1: Add statistical significance of differences in receptor expression between studies. 

We appreciate the suggestion. Unfortunately, direct statistical comparison is not feasible due to fundamental differences in the normalization methods used: FPKM accounts for transcript length and sequencing depth, while TMM-normalized log2(CPM) adjusts for compositional differences between libraries but does not correct for transcript length. As a result, the expression values are on different scales and not directly comparable. Nevertheless, Table A1 provides an overview of receptor detectability across datasets. 

Round 3

Reviewer 1 Report

Comments and Suggestions for Authors

Thanks for the answers and corrections. The authors took into account all my comments. The article can be accepted in its current form.